# Role of m6A RNA Methylation in Thyroid Cancer Cell Lines

**DOI:** 10.3390/ijms231911516

**Published:** 2022-09-29

**Authors:** Lorenzo Allegri, Federica Baldan, Elisabetta Molteni, Catia Mio, Giuseppe Damante

**Affiliations:** 1Department of Medicine, University of Udine, Via Chiusaforte, 33100 Udine, Italy; 2Institute of Medical Genetics, Academic Hospital of Udine, Azienda Sanitaria Universitaria Integrata di Udine, 33100 Udine, Italy

**Keywords:** RNA methylation, m6A, METTL3, thyroid cancer, epitranscriptome

## Abstract

N6-methyladenosine (m6A) is the most abundant internal modification of RNA in eukaryotic cells, and, in recent years, it has gained increasing attention. A good amount of data support the involvement of m6A modification in tumorigenesis, tumor progression, and metastatic dissemination. However, the role of this RNA modification in thyroid cancer still remains poorly investigated. In this study, m6A-related RNA methylation profiles are compared between a normal thyroid cell line and different thyroid cancer cell lines. With this approach, it was possible to identify the different patterns of m6A modification in different thyroid cancer models. Furthermore, by silencing METTL3, which is the main player in the RNA methylation machinery, it was possible to evaluate the impact of m6A modification on gene expression in an anaplastic thyroid cancer model. This experimental approach allowed us to identify *DDI2* as a gene specifically controlled by the m6A modification in anaplastic thyroid cancer cell lines. Altogether, these data are a proof of concept that RNA methylation widely occurs in thyroid cancer cell models and open a way forward in the search for new molecular patterns for diagnostic discrimination between benign and malignant lesions.

## 1. Introduction

Thyroid cancer represents the most prevalent type of endocrine malignancy, accounting for 1–2% of all cancer cases worldwide [1,2]. Most thyroid carcinomas derive from follicular cells and are classified according to their differentiation levels in differentiated thyroid cancer, including papillary thyroid cancer (PTC) and follicular thyroid cancer (FTC), or in poorly differentiated thyroid cancer or anaplastic thyroid cancer (ATC)^3^. PTC and FTC account for more than 95% of thyroid tumors, but despite their high incidence, they are the most responsive to the current treatment based on surgery and following radioiodine therapy, resulting in an overall survival rate of over 90% within 10 years. ATC is a rare but aggressive form of thyroid malignancy. Indeed, despite the fact that it only accounts for <2% of all thyroid malignancies, it is responsible for 20–50% of thyroid cancer-associated deaths [1,2]. It is mostly diagnosed as stage IV disease [3], often not resectable, and not responsive to radioactive iodine ablation for its undifferentiated phenotype due to the loss of thyroid-specific gene expression [4]. The current management of ATC has strongly been conditioned by the advent of the molecular testing of thyroid tumors that allows targeting the known genetic mutations [5] with the new molecular drugs, mainly inhibitors of tyrosine kinases [6,7]. However, for the tumors resistant to these new drugs and those without known detectable genetic aberrations, the current treatment is still represented by palliative surgery and systemic chemotherapy [8]. Unfortunately, even combination therapies easily induce drug resistance, sentencing ATC-burdened patients to death due to unlimited local growth and distal metastasis dissemination [8,9]. For these reasons, a better understanding of the molecular mechanisms that regulate tumorigenesis, tumor progression, and treatment-related resistance is crucial for the development of early detection biomarkers and novel therapeutic strategies.

Among numerous RNA modifications, N6-methyladenosine (m6A) is the most abundant one with an average of 1–2 m6A residues every 1000 nucleotides [10,11]. m6A can be reversibly added and removed in a fluid pattern that makes it an actual epigenetic modification [12,13]. m6A alters target gene expression, thus influencing the corresponding cell processes and physiological functions. Furthermore, m6A participates in almost all steps of RNA metabolism, including mRNA translation, degradation, splicing, export, and folding [14,15]. m6A modification is mediated by three classes of enzymes: (i) methylators, which include the m6A methyltransferases methyltransferase-like 3 (METTL3), the methyltransferase-like 14 (METTL14), and Wilms tumor 1 associated protein (WTAP); (ii) erasers, comprising the fat-mass and obesity-associated protein (FTO) and the alkylation repair homolog protein 5 (ALKBH5) [16,17,18,19]; (iii) readers, which include the YT521-B homology (YTH) domain family (YTHDF1-3 and YTHDC1-2), insulin-like growth factor 2 mRNA-binding proteins (IGF2BP1–IGF2BP3), and eukaryotic initiation factor 3 (EIF3) [20]. METTL3 is an S-adenosyl methionine (SAM)-binding protein highly conserved in eukaryotes and is the most important component of the m6A methyltransferase complex (MTC) [21]. METTL3 and METTL14 are colocalized in nuclear speckles and form stable complexes in a 1:1 ratio. However, only METTL3 acts as a methylator by its internal SAM-binding domain that catalyzes the transfer of methyl groups in SAM to adenine bases in RNA, producing S-adenosyl homocysteine (SAH). On the other hand, METTL14 primarily acts to stabilize the structure of MTC, and it identifies specific RNA sequences (“RRACH”) as catalytic substrates [22,23]. Many studies have demonstrated an active role of m6A in tumorigenesis and tumor progression, i.e., through the regulation of mRNA methylation of oncogenes or tumor suppressors. The different methylation pattern leads to a fine-tuning of the interaction between target mRNAs and downstream readers, which in turn leads to a regulation of the biological processes that sustain cancer cells [24]. Although many studies have established a close relationship between METTL3 expression or m6A signature and malignant phenotypes, the potential role of METTL3 and m6A RNA modification in thyroid cancer is still largely elusive.

## 2. Results

### 2.1. m6A Patterns in Thyroid Cells

To date, mRNA methylation patterns and qualitative distribution of m6A in the transcriptome of normal and thyroid cancer cell lines have not been yet described. For this purpose, in a first experimental setting, RNA extracted from a human normal thyroid cell line (Nthy-ori-3.1) and three thyroid cancer cell lines (BCPAP, SW1736, and 8505C) was immunoprecipitated by a specific m6A antibody. Immunoprecipitated RNA samples were, then, subjected to sequencing using a meRIP-seq protocol in order to identify m6A-enriched mRNAs. In the high-throughput meRIP-sequencing analysis, we identified 4672 transcripts significantly m6A-enriched in Nthy-ori-3.1, 6296 in BCPAP, 5134 in SW1736, and 1626 in 8505C cells. Among them, 710 transcripts were specifically enriched only in Nthy-ori-3.1, 1758 in BCPAP, 1115 in SW1736, and 97 in 8505C. Considering anaplastic thyroid cancer cells vs. normal thyroid cells, 74 targets were commonly enriched in both ATC cell lines (SW1736 and 8505C). Sequencing data are summarized in Table 1.

Taking into account the totality of m6A-enriched mRNAs, the principal component analysis (PCA), revealed a clear separation between Nthy-ori-3.1, BCPAP, and SW1736 cells (Figure 1a), indicating heterogeneity among the four cell lines. Unexpectedly, Nthy-ori-3.1 and 8505C cells are relatively close together. This could be partially explained by the fact that 8505C retains partial features of a well-differentiated carcinoma, whereas SW1736 cells are completely de-differentiated; this spatially places them between Nthy-ori-3.1 and SW1736 in the PCA. These data were also confirmed with a heat map in which it can be clearly seen that 8505C is markedly different from the other two tumor cell lines, probably due to the low number of enriched mRNAs (Figure 1b). Altogether, these data indicate a significant heterogeneity of m6A-enriched mRNAs among the four cell lines.

Table 2 lists the top 20 mRNAs most enriched in m6A for each cell line. While most of them are specific for a single cell line or at least present in two out of four, only *U2AF1* is present in three of the four lines.

After analyzing the data obtained by meRIP-seq in the individual cell lines, the m6A-enriched mRNAs in each cell line were subjected to ontology-based pathway analysis in order to outline which pathways were mostly involved in m6A addition in each cell line. We selected pathways in which at least two components were enriched in the selected model (Figure 2). With these settings, five pathways are common for all four cell lines (Wnt, CCKR, gonadotropin-releasing hormone receptor (GNRH) signaling, angiogenesis, and apoptosis). This suggests that m6A modification of mRNAs involved in these processes could play a key and not coincidental role in their modulation. Interestingly, though Nthy-ori-3.1 and 8505C cells are close in the PCA, only the TGF-beta signaling pathway is unique and common. Seven pathways are shared between the two anaplastic cell lines.

### 2.2. Effects of METTL3 Silencing in ATC Cell Lines

Once the different profiles of m6A in the four thyroid cell lines were analyzed, the effect of mRNA methylation in terms of cell proliferation was assessed. For this purpose, we down-regulated *METTL3* expression by RNA interference in the two ATC cell lines. Even though m6A methylation is catalyzed by a multicomponent methyltransferase complex [25], METTL3 primarily functions as the catalytic core and consequently is indispensable for the addition of methylation. After confirming METTL3 protein silencing (Figure 3a,b), we estimated the effects of its silencing on cell viability in ATC cell lines by performing an MTT assay. As shown in Figure 3c, METTL3 silencing induces a strong and significant cell viability reduction, compared to the negative control. To obtain further confirmation about the effects on cell viability and proliferation, cells were counted after the addition of trypan blue. The results obtained strongly overlapped with those observed after the MTT assay (Appendix A). These results suggest that METTL3 could play a central role in ATC cell proliferation and survival. In order to confirm the effects of METTL3 down-regulation on m6A methylation deposition on mRNAs, total m6A levels in ATC-extracted mRNAs after silencing were measured. As shown in Figure 3d, the reduction in METTL3 induced by the two siRNAs results in a significant reduction in total m6A levels, confirming that METTL3 silencing is a proper strategy for altering the mechanism of m6A methylation in these cell lines.

### 2.3. METTL3 Silencing Effects on Gene Expression 

To identify the molecular mechanism behind the effects observed in ATC cell lines after METTL3 silencing, we performed a high-throughput RNA sequencing analysis in SW1736 and 8505C cells. To assess the transcriptional changes induced by the down-regulation of *METTL3* expression, a comparison between cells treated with negative control and cells treated with siRNA 1 or siRNA 2 was performed. Considering these data, coupled with the principal component analysis and the heat maps shown in Figure 4 (panels a–b), it is clear that the two cell lines possess a diverse transcriptome. When METTL3 is silenced in SW1736, a strong and general reduction in transcription is induced, whereas, in 8505C, the observed effect appears to be less evident (Figure 4a) and in the opposite direction (Figure 4b).

Merging data from siRNAs 1 and 2 in each cell lines, 389 targets were commonly up-regulated and 379 targets were down-regulated in SW1736, while 616 targets were commonly up-regulated and 271 were down-regulated in 8505C. The mean between the two siRNAs was calculated. Only mRNAs with a log 2 fold change less than −2.5 or greater than 2.5 were considered. As for commonly deregulated genes in the two ATC cell lines, 63 up-regulated and 39 down-regulated RNAs were obtained (Figure 5a) The RNAs commonly up- or down-regulated in ATC cell lines were, then, subjected to ontology-based pathway analysis. With regard to up-regulated genes, the pathways most involved seem to be: integrin signaling; heterotrimeric G-protein signaling pathway, Gi alpha and Gs alpha mediated; and PDGF signaling pathway (Figure 5b). On the other hand, analyzing the commonly down-regulated genes, the only pathway with more than one gene involved was the gonadotropin-releasing hormone receptor (GRHR) pathway (Figure 5c).

### 2.4. Relations between meRIP-seq and Effects of METTL3 Silencing on Gene Transcription

In the final phase of this study, the relationship between m6A-enriched mRNAs and their transcript levels was evaluated. As demonstrated, silencing of *METTL3* results in a marked and general reduction in RNA methylation levels in addition to the change in transcription of thousands of genes. Data from meRIP-seq analysis and those obtained by RNA-seq following *METTL3* silencing were cross-referenced. Table 3 shows the number of m6A-enriched RNAs and up- or down-regulated mRNAs after *METTL3* silencing in the two cell lines, both for the single cell lines or crossing data from the two cell lines. Considering the thousands of genes enriched and the thousands of genes for which transcript levels varied after *METTL3* silencing, the reduced order of magnitude of the number of genes both m6A-enriched and deregulated after RNA interference suggests that for these target mRNAs there is a close link between methylation and expression regulation.

Focusing on the common genes in the two cell lines, only six up-regulated genes and one down-regulated gene were assessed, confirming the clear diversity between the two cell lines. Commonly deregulated genes are listed in Table 4.

Assuming a direct relationship between addition of the m6A modification to the mRNA and the expression of that particular gene, we focused on the only gene that was found to be down-regulated as a result of *METTL3* silencing. The expression levels of *DDI2* were reassessed by real-time PCR, confirming the significant reduction in *DDI2* after treatment with both siRNAs and in both ATC cell lines (Figure 6a). To further investigate the extent of the effects of *METTL3* silencing on *DDI2*, its protein expression levels were assessed. Figure 6 (panels b–c) shows that DDI2 protein levels are also significantly reduced for at least one siRNA in both cell lines. Indeed, this represents a strong indication of the fact that the expression of DDI2 is finely regulated by the addition of m6A to its mRNA sequence.

## 3. Discussion

In recent years, evidence for the relevant role of m6A in cancer development, progression, and aggressiveness has been increasingly gained. Recently, many studies have chosen one or two m6A regulators to explore the aberrant expression and underlying mechanism of m6A in cancer. In this way, the role of various actors involved in m6A addition, reduction, or readout in numerous types of neoplasia could be demonstrated [26,27]. Most of these studies have investigated this phenomenon by focusing, for example, on METTL3 and one or a few genes whose expression is up- or down-regulated by the addition or removal of the m6A modification on their mRNA [28], whereas works studying the effects of the modification of the entire m6A pattern are much more complicated and consequently less frequent. In this study, for the first time, m6A RNA methylation profiles in four different thyroid cell lines are compared. Principal component analysis (PCA) was performed and heat maps were generated considering total m6A-enriched mRNAs, which revealed a clear separation of normal thyroid cells (Nthy-ori-3.1) and cancer ones (BCPAP, SW1736, and 8505C) (Figure 1). In particular, there is a marked difference between ATC and PTC cells (i.e., BCPAP), suggesting very different patterns of m6A between differentiated and undifferentiated thyroid cancer. Such diversity in m6A profiles among cells representing healthy tissue models, moderately aggressive tumors, and highly aggressive tumors could act as a springboard for developing new approaches to refine the diagnosis of thyroid cancers. Nowadays, in fact, a certain level of uncertainty remains when fine-needle aspirate biopsy (FNAB) produces diagnostic category 3 or 4 as an outcome [29].

For the first time, the main pathways whose involved genes are found to be enriched in m6A in different thyroid cell lines or subgroups of them are here reported (Figure 2). This finding suggests the role that RNA methylation plays in normal cells, or in cancer cells with different degrees of aggressiveness. These results, although extremely preliminary, shed light on a possible m6A-dependent approach in the study of new diagnostic methods or in the identification of target pathways in the treatment of thyroid cancer.

A small subset of thyroid cancers, known as ATC, is undifferentiated and nearly incurable, with a median survival of only six months. Because of its dismal prognosis, it is responsible for 40–50% of total thyroid cancer-related deaths [30]. In recent years, research has produced great efforts in an attempt to find an effective treatment against ATC, such as the use of taxane, dabrafenib, or trametinib (in patients harboring the BRAF p.V600E mutation) [31]; natural compounds such as DHT [32]; or molecules that can alter epigenetic pathways [33]. However, there is still no standardized and effective treatment for this type of malignancy, partly because ATC cells are able to evade most therapeutic strategies by developing chemoresistance [32]. For these reasons, we focused on analyzing the effects of m6A in ATC cells. To the best of our knowledge, this study investigates for the first time the effects of *METTL3* in an ATC model. Data about the role of *METTL3* in thyroid cancer are few and elusive, but for the most part, they suggest an anticancer-like action [34,35]. Instead, the results of this study demonstrate how *METTL3* down-regulation results in decreased ATC cell viability (Figure 3), suggesting how this component of the RNA methylation machinery plays a favorable role in the development, progression, and maintenance of anaplastic thyroid cancer. This result is consistent with others that identify other epigenetic regulatory mechanisms involving mRNAs (both in terms of modifications and interactions) downstream of gene transcription as particularly relevant in the regulation of ATC [36]. In this study, not only the effects of *METTL3* silencing on ATC cell line viability but also its effects on the transcriptome (Figure 4 and Figure 5) are investigated for the first time. The two cell lines used in this study, despite both representing a model of ATC, showed a large difference in both the pattern of m6A and the effects of *METTL3* silencing on the transcriptome. 8505C cells showed less alteration in terms of gene expression following the silencing of METTL3. The most consistent hypothesis capable of explaining what was observed concerns the fact that this cell line exhibited extremely low m6A enrichment, which suggests that given the small number of mRNAs bearing such an alteration, switching off their methylator changes only the small proportion of methylated mRNAs. Considering this intrinsic difference and that genes presenting m6A modification and whose levels were similarly modified by *METTL3* down-regulation in both cell lines are extremely few, the obtained results highlight that these targets are the core effectors of m6A methylation in this model. Given the already complex nature of this study, we focused only on the m6A-presenting and down-regulated genes in both cell lines after *METTL3* silencing, verifying that only DDI2 exhibits these features (Figure 6). The molecular role of DDI2 in thyroid cancer is almost unknown. Tomei and colleagues [37] have shown that this gene in combination with nine others can provide useful diagnostic information to establish the malignant or benign nature of thyroid lesions after biopsy. A more recent study suggests that DDI2 plays a pro-tumoral role by activating NRF1 and ultimately increasing the ability of cancer cells to degrade ubiquitinated proteins in a proteosome-dependent manner. Both of these hypotheses could support a relevant role of DDI2 both as a biomarker and as one of the players involved in the aggressiveness of ATCs, compared with healthy tissue or more differentiated thyroid tumors. Overall, this study, although carried out on continuous cell models, provides for the first time a great deal of new information about the characteristics peculiar to healthy, cancerous, and aggressively cancerous thyroid models. Although the data presented here are extremely preliminary, they open a way forward in the search for new molecular patterns for diagnostic discrimination between benign and malignant lesions, in the study of the role of m6A in thyroid cancer, and in the search for new druggable pathways in subtypes of thyroid tumors that, to date, are incurable. Techniques for analyzing m6A profiles currently require large amounts of mRNA, which has prevented us from performing meRIP-seq after *METTL3* silencing, both because the cell lines used show a sharp decrease in cell viability after treatment and because a marked decrease in total m6A level is observed. These two drawbacks make the technical conditions for conducting such an experiment prohibitive. However, we are confident that the refinement of meRIP techniques and the emergence of different approaches [38] will soon allow not only the direct analysis of the changes in m6A profiles after *METTL3* down-regulation, but also the application of this approach to tissue samples. 

## 4. Materials and Methods

### 4.1. Cell Lines

In this study, we used four different thyroid cell lines: Nthy-ori-3.1, derived from normal thyroid follicular epithelial cells and immortalized by the SV40 large T antigen; BCPAP, derived from papillary thyroid carcinoma; SW1736 and 8505C, from anaplastic thyroid cancer. All cell lines have been validated by short tandem repeat and tested for being mycoplasma-free. Cells were grown in RPMI 1640 medium (EuroClone, Milan, Italy) supplemented with 10% fetal bovine serum (Gibco Invitrogen, Milan, Italy), 2 mM L-glutamine (EuroClone, Milan, Italy), and 50 mg/mL gentamicin (Gibco Invitrogen, Milan, Italy). Cells were maintained in a humidified incubator (5% CO_2_, 37 °C). 

### 4.2. Methylated RNA Immunoprecipitation

Methylated RNA Immunoprecipitation (meRIP) was performed using the Magna MeRIP m6A Kit (Millipore, Burlington, MA, USA) according to the manufacturer’s instructions. In order to perform the MeRIP assay, total RNA from Nthy-ory-3.1, BCPAP, SW1736, and 8505C cell lines was extracted with an RNeasy mini kit according to the manufacturer’s instructions (Qiagen, Hilden, Germany). Three hundred micrograms per condition of total RNA was incubated overnight with Protein A/G Magnetic Beads complexed with rabbit polyclonal anti-m6A RIP+ antibody (Millipore, Burlington, MA, USA) or normal Rabbit IgG (Millipore, Burlington, MA, USA) as a negative control. The day after, immune complexes were eluted and the RNA was column purified. RNA obtained from meRIP was sequenced and analyzed as described below.

### 4.3. meRIP-seq Library Preparation and Sequencing

TruSeq Stranded Total RNA with Ribo-Zero Human/Mouse/Rat kit (Illumina, San Diego, CA, USA) was used for library preparation following the manufacturer’s instructions, starting with 200 ng of good-quality RNA (R.I.N. > 7) as input. Both RNA samples and final libraries were quantified by using the Qubit 2.0 Fluorometer (Invitrogen, Waltham, MA, USA) and quality tested by Agilent 2100 Bioanalyzer RNA Nano assay (Agilent technologies, Santa Clara, CA, USA). Libraries were then processed with Illumina cBot for cluster generation on the flowcell, following the manufacturer’s instructions, and sequenced in 50 bp single-end mode on the HiSeq2500 (Illumina, San Diego, CA, USA). The CASAVA 1.8.2 version of the Illumina pipeline was used to process raw data for both format conversion and de-multiplexing.

### 4.4. meRIP-Seq Bioinformatics Analysis

Raw sequence files were subjected to quality control analysis using FastQC (http://www.bioinformatics.babraham.ac.uk/projects/fastqc/, accessed on 5 March 2021). In order to avoid low-quality data, adapters were removed using Cutadapt [39] and lower-quality bases were trimmed using ERNE. For the analysis of differentially expressed genes, the quality-checked reads were processed using the TopHat version 2.0.0 package (Bowtie 2 version 2.2.0) as FASTQ files. The reads were mapped to the human reference genome GRCh37/hg19. Read abundance was evaluated and normalized by using Cufflinks [40] for each gene, and fragments per kilobase of exon per million fragments mapped (FPKM) values were obtained. Cuffdiff from the Cufflinks 2.2.0 package was used to calculate the differential expression levels and to evaluate the statistical significance of detected alterations between control and silenced samples, in both cell lines. For further analysis, we selected effective data using three criteria: (a) FPKM values > 0.5; (b) log 2 fold change > 1.5 and (c) q-value (false discovery rate (FDR)) < 0.005. PCA analysis was performed with the free online tool for principal components analysis Clustvis [41]; the heatmaps were obtained by processing our sequencing data with the online tool Morpheus (https://software.broadinstitute.org/morpheus, accessed on 11 March 2021).

### 4.5. RNA Extraction and High-throughput Sequencing

Total RNA from SW1736 and 8505C cell lines was extracted as previously described, and the Qubit RNA HS assay (ThermoFisher Scientific, Waltham, MA, USA) was used to quantify RNA on a Qubit 4.0 Fluorometer. cDNA was generated using the SuperScript IV VILO Kit (ThermoFisher Scientific, Waltham, MA, USA). Barcoded libraries were prepared using the Ion AmpliSeq Transcriptome Panel Human Gene Expression CORE (Thermo Fisher Scientific, Waltham, MA, USA) and the Ion AmpliSeq Library Kit Plus (Thermo Fisher Scientific, Waltham, MA, USA), following the manufacturer’s protocol. All the reactions were performed in a Veriti Dx 96-Well Thermal Cycler (Applied Biosystems, Waltham, MA, USA). Barcoded libraries were quantified with the Qubit dsDNA HS Assay kit (Life Technologies, Carlsbad, CA, USA) and then diluted to 100 pM. Libraries were loaded into the Ion Chef instrument (Thermo Fisher Scientific, Waltham, MA, USA) for template enrichment and chip loading. Sequencing was performed with the Ion S5 GeneStudio Sequencer using the Ion 540 Kit-Chef and the Ion 540 chip-kit (all Thermo Fisher Scientific, Waltham, MA, USA). Reads were aligned to the reference genome and the RNA-seq analysis plugin was run on the Torrent Suite Server (Thermo Fisher Scientific, Waltham, MA, USA). Gene isoform representations were extracted, and comparative representation plots across barcodes were created in addition to individual reports for each barcode.

### 4.6. METTL3 Silencing

For transient silencing of endogenous *METTL3*, TriFECTa RNAi Kit (Integrated DNA Technologies Inc, Coralville, IA, USA) was used. A “universal” negative control duplex, which targets a site absent in the human genome, was used. Two different siRNA oligonucleotides (siRNA 1 and siRNA 2) were transfected at a concentration of 5 nM using DharmaFECT 1 Transfection reagent (ThermoFisher Scientific), according to the manufacturer’s instructions. The day before transfection, SW1736 and 8505C cells were plated in an antibiotic-free medium. Cells were harvested 48 h after transfection, and gene-silencing efficiency was evaluated by protein level analysis. 

### 4.7. Total m6A Abundance Assay

Total m6A abundance was evaluated by using EpiQuik m6A RNA Methylation quantification kit (Epigentek, Farmingdale, NY, USA). Briefly, 200 ng of total RNA per condition was bound to assay wells by incubation at 37 °C for 90 minutes and then incubated with the m6A capture antibody and detection antibody at room temperature for 30 minutes. After incubation with color developing solution, the absorbance was detected on a microplate reader at 450 nM. 

### 4.8. Cell Viability

In order to test cell viability, we applied the methylthiazolyldiphenyl-tetrazolium bromide (MTT) assay. SW1736 and 8505C cells (3000 cells/well) were plated onto 96-well plates in 200 μL medium/well and were allowed to attach to the plate for 24 h (t0). Plates were then treated either with negative control (nc) or with two *METTL3*-specific siRNAs (siRNA 1 and siRNA 2) for 72 hours. Then, 4 mg/mL MTT (Sigma-Aldrich) was added to the cell medium, and cells were cultivated for another 4 hours in the incubator. The supernatant was removed, 100 μL/well of DMSO (Sigma-Aldrich) was added, and the absorbance at 570 nM was measured. All experiments were run sixfold and cell viability was expressed as a fold change compared to control. To confirm cell viability data, 6 × 10^6^ cells were plated and left to grow for 24 hours, after which they were treated with negative control or METTL3-specific siRNAs. After 72 hours of treatment, cells were detached, stained with trypan blue, and counted under a light microscope.

### 4.9. Target Gene Expression Assay

A total of 500 ng total RNA from SW1736 and 8505C cells was extracted as described above and reverse transcribed to cDNA using random hexaprimers and SuperScript IV reverse transcriptase (Thermo Fisher Scientific, Waltham, MA, USA). Quantitative PCR was performed using PowerUP Sybr green master mix (Thermo Fisher Scientific, Waltham, MA, USA) on the QuantStudio3 system (Applied Biosystems, Waltham, MA, USA). The QuantStudio Design and Analysis software v1.5.0 (Applied Biosystems), was used to calculate mRNA levels with the 2^−ΔΔ^Ct method, and ß-actin was used as a reference. All experiments were performed in triplicate. Oligonucleotide primers were purchased from Sigma-Aldrich, and the sequences of the primers are listed in the Appendix A.

### 4.10. Protein Extraction and Western Blot

Briefly, SW1736 and 8505C cells were harvested by scraping and lysed with total lysis buffer (Tris HCl 50 mM pH8, NaCl 120 mM, EDTA 5 mM, Triton 1%, NP40 1%, protease inhibitors). Lysates were centrifuged at 13,000× *g* for 10 min at 4 °C, and supernatants were quantified using the Bradford assay. For Western blot analysis, proteins were electrophoresed on 10% SDS-PAGE and then transferred to nitrocellulose membranes, saturated with 5% non-fat dry milk in PBS/0.1% Tween 20. The membranes were then incubated overnight with mouse monoclonal anti-DDI2 antibody 1:500 (Santa Cruz Biotechnology), rabbit monoclonal anti-METTL3 antibody 1.1000 (Abcam, Cambridge, UK), or rabbit anti-β-actin antibody 1:1000 (Abcam, Cambridge, UK). The day after, membranes were incubated for 2 h with anti-rabbit immunoglobulin coupled to peroxidase 1:4000 (Sigma-Aldrich, St. Louis, MO, USA). Blots were developed using UVITEC Alliance LD (UVITec Limited, Cambridge, UK) with SuperSignal Technology (Thermo Fisher Scientific, Waltham, MA, USA).

### 4.11. Statistical Analysis

All data were expressed as means ± SD, and significances were analyzed with either Student’s t-test or one-way ANOVA, both performed using GraphPad Prism (GraphPad Software, Inc., San Diego, CA, USA).

## Figures and Tables

**Figure 1 ijms-23-11516-f001:**
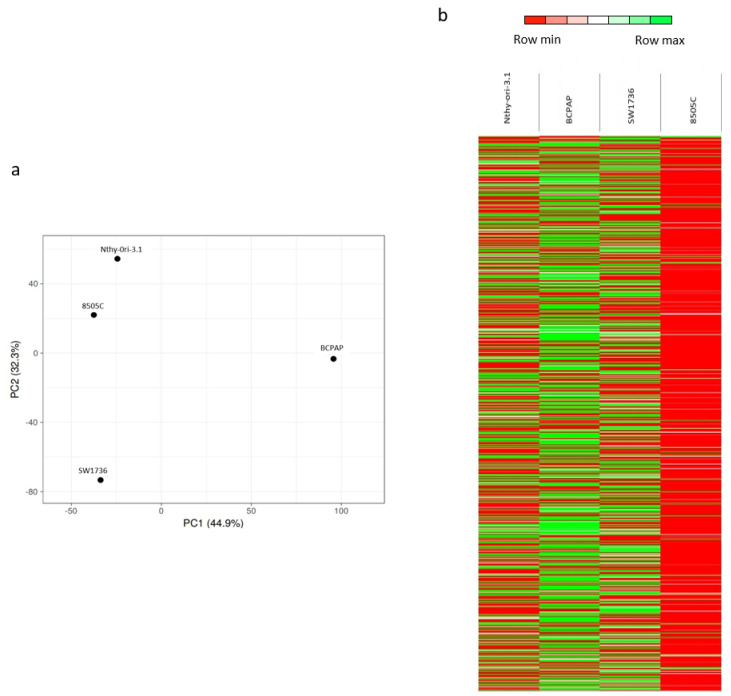
m6A patterns in thyroid normal and cancer cells. (**a**) Principal component analysis (PCA) results. Each point represents an meRIP-seq sample. Samples with similar gene expression profiles clustered together. (**b**) Heat maps showing the hierarchical clustering of m6A-enriched mRNAs in Nthy-ori-3.1, BCPAP, SW1736, and 8505C. Results are shown as fold change (log 2).

**Figure 2 ijms-23-11516-f002:**
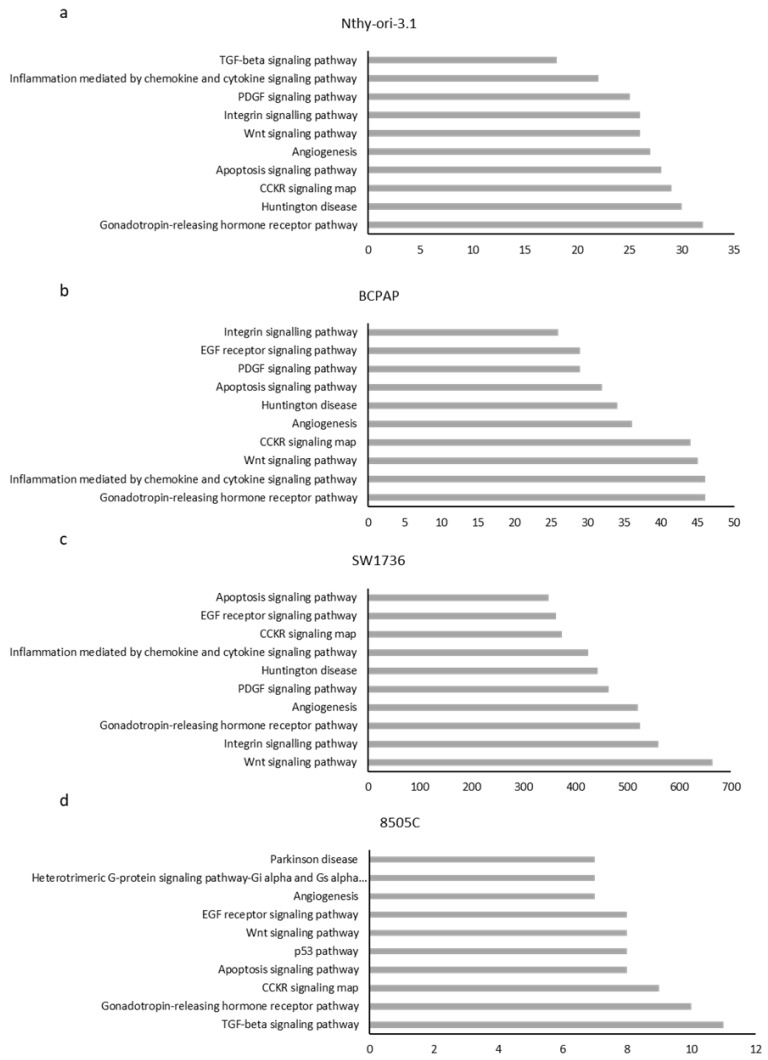
Histogram representing the top 10 pathways related to m6A enrichment in (**a**) Nthy-ori-3.1, (**b**) BCPAP, (**c**) SW1736, and (**d**) 8505C cell lines. The x-axis shows the number of genes involved in each pathway.

**Figure 3 ijms-23-11516-f003:**
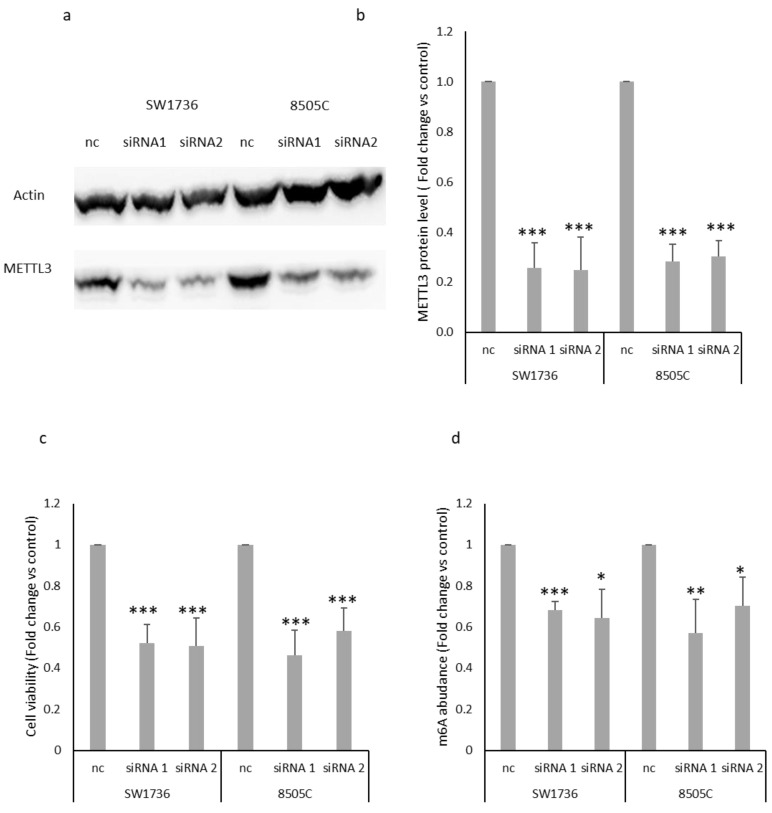
Effects of METTL3 silencing on cell viability and m6A total methylation in ATC cells. (**a**) Blot image of SW1736 and 8505C cells after 72 h transfection with non-targeting siRNA (nc, negative control) or two different siRNAs (siRNA 1 and siRNA 2) specific to *METTL3*. (**b**) Densitometric analysis of METTL3 protein levels in ATC cells treated with negative control (nc) or two different siRNAs (siRNA 1 and siRNA 2) specific to METTL3. Data are normalized against β-actin levels. Nc was set to 1 and protein levels were expressed as fold change versus control. *n* = 3. *** *p* < 0.001. (**c**) Cell viability of SW1736 and 8505C cells, transfected with two siRNAs (siRNA 1 and siRNA 2) or negative control (nc) for 72 h, was analyzed by MTT assay. nc was set at 1 and cell viability was expressed as fold change versus control. *n* = 6. *** *p* < 0.001. (**d**) Colorimetric quantification of N6-methyladenosine in mRNA in ATC cells after *METTL3* silencing (siRNA 1 and siRNA 2) or in negative control (nc). Nc was set to 1 and m6A RNA abundance was expressed as fold change versus control * *p* < 0.05, ** *p* < 0.01, *** *p* < 0.001.

**Figure 4 ijms-23-11516-f004:**
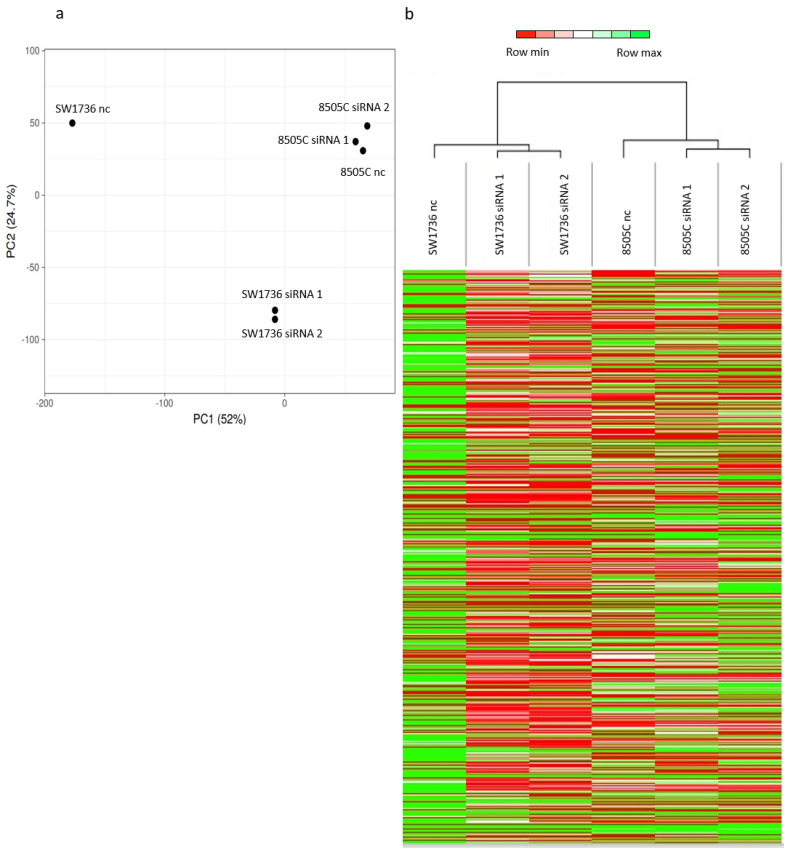
Gene expression analysis after METTL3 silencing in ATC cells. (**a**) Principal component analysis (PCA) results. Each point represents an RNA-seq sample. Samples with similar gene expression profiles are clustered together. (**b**) Heat maps showing the hierarchical clustering of RNAs in untreated cells (SW1736 nc/8505C nc) and in METTL3 siRNA-treated ones (SW1736 siRNA 1/8505C siRNA 1 and SW1736 siRNA 2/8505C siRNA 2) and 8505C. Results are shown as fold change (log 2).

**Figure 5 ijms-23-11516-f005:**
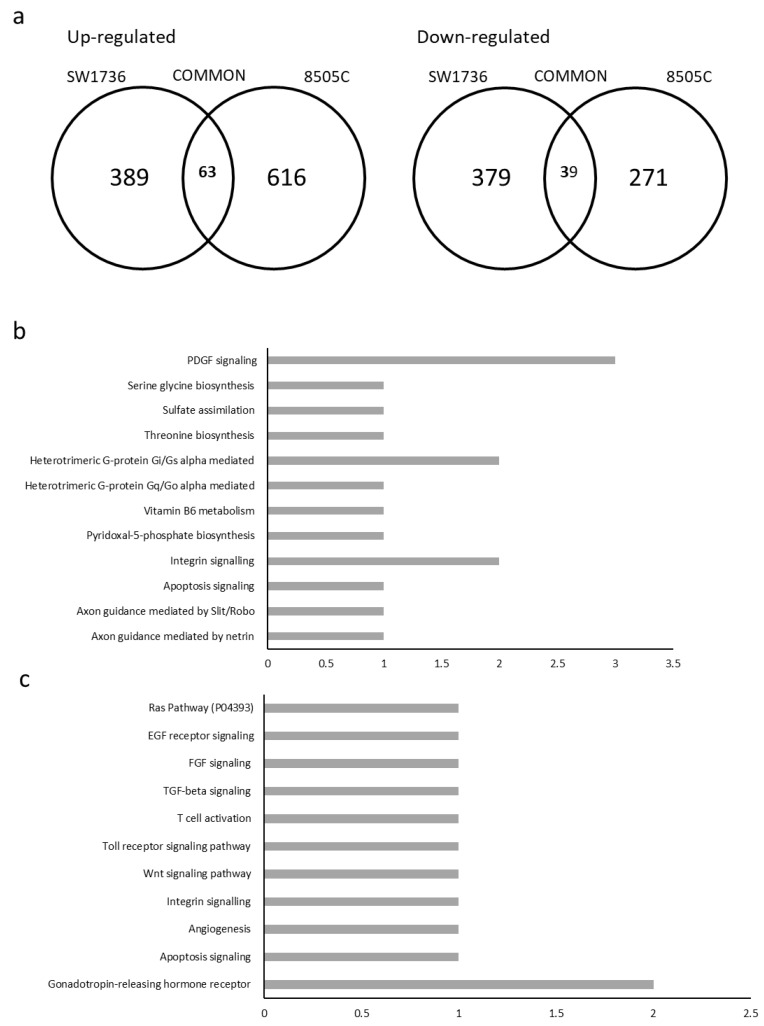
Pathway analysis after *METTL3* silencing. (**a**) Number of genes up-regulated (left) or down-regulated (right) in each cell line or common between the two cell lines (intersection of the sets). List of pathways and the number of genes involved in each for up-regulated (**b**) and down-regulated (**c**) genes. The x-axis shows the number of genes involved in each pathway.

**Figure 6 ijms-23-11516-f006:**
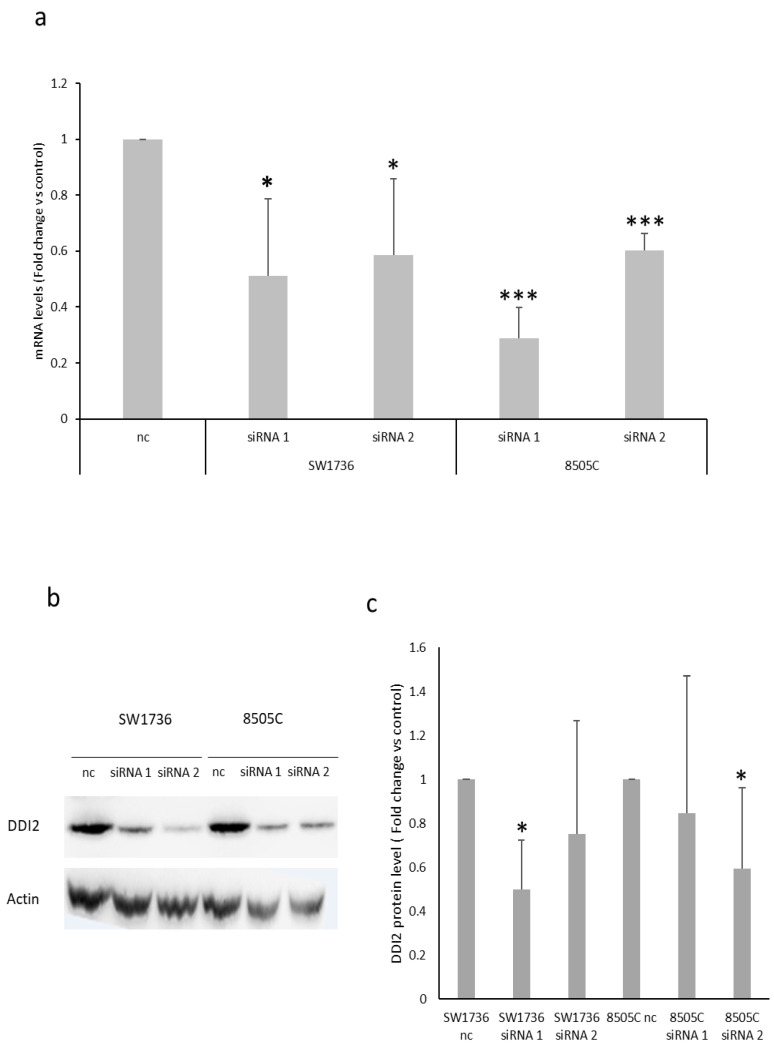
Investigation of targets commonly down-regulated after METTL3 silencing in ATC cells. (**a**) mRNA levels for *DDI2* in ATC cell lines treated with negative control (nc) or *METTL3*-specific siRNAs (siRNA 1 or siRNA 2) for 72h. Nc was set to 1 and protein levels were expressed as fold change versus control. *n* = 3. * *p* < 0.05, *** *p* < 0.001. (**b**) Blot image of SW1736 and 8505C cells after 72 h transfection with non-targeting siRNA (nc, negative control) or two different siRNAs (siRNA 1 and siRNA 2) specific to *METTL3*. (**c**) Densitometric analysis of DDI2 protein levels in ATC cells treated with negative control (nc) or two different siRNAs (siRNA 1 and siRNA 2) specific to *METTL3*. Data are normalized against β-actin levels. Nc was set to 1 and protein levels were expressed as fold change versus control. *n* = 5. * *p* < 0.05.

**Table 1 ijms-23-11516-t001:** Numbers of total and cancer-specific mRNAs containing m6A (total m6A-enriched mRNAs and specific m6A-enriched mRNAs, respectively).

Cell Line	Total m6A-Enriched mRNAs	Specific m6A-Enriched mRNAs
Nthy-ori-3.1	4672	710
BCPAP	6296	1758
SW1736	5134	1115
8505C	1626	97
ATC	1286	74

**Table 2 ijms-23-11516-t002:** Top 20 m6A-enriched mRNAs for Nthy-ori-3.1, BCPAP, SW1736, and 8505C cell lines.

Nthy-ori-3.1		BCPAP		SW1736		8505C	
mRNA	log 2 (m6A/IgG)	mRNA	log 2 (m6A/IgG)	mRNA	log 2 (m6A/IgG)	mRNA	log 2 (m6A/IgG)
MRPS24	23.5	VKORC1	23.4	NDUFB10	20.8	CBS	17.9
SAT2	22.9	NDUFA7	23.4	COX5B	20.5	EMC6	17.8
APRT	22.8	SPANXB1	23.2	MRPL40	20.3	UTP14C	17.3
RNF181	22.2	NCBP2	22.9	MRPS24	20.0	PILRB	17.2
KCTD14	22.2	SWI5	22.9	MRPL14	19.8	RDH14	17.1
MRPL52	22.1	C12orf45	22.6	CD3EAP	19.5	NPIPA5	16.7
U2AF1	21.9	HNRNPH2	22.6	SDF2L1	19.4	SAPCD1	16.2
ZNF593	21.9	IFI6	22.6	NR2F6	19.4	SMIM11	16.0
MED11	21.8	AAMDC	22.4	RPP21	19.4	PPAN	15.9
SURF2	21.7	MYL6B	22.3	MRPS17	19.3	C21orf33	15.7
PDF	21.6	SURF2	22.3	PALM2	19.2	ICAM2	15.6
PAM16	21.6	DUSP23	22.3	APRT	19.0	NPIPA2	15.6
DUS1L	21.5	TRPT1	22.2	MND1	19.0	GPR89B	15.5
MRPL55	21.5	NKAP	22.2	MITD1	18.9	C9orf116	15.5
SSNA1	21.4	C17orf49	22.2	ISOC2	18.9	TRIM34	15.5
POLR2M	21.4	MED11	22.1	GPKOW	18.9	C1orf53	15.4
NUPR1	21.4	PSMB8	22.1	U2AF1	18.9	DNLZ	15.3
CSNK1E	21.3	RP9	22.0	FKBP11	18.8	GSTT2B	15.3
SDF2	21.3	U2AF1	21.9	FDX1L	18.7	LIN7B	15.2
FDX1L	21.3	EAPP	21.9	MRPS12	18.7	FAM156A	15.2

**Table 3 ijms-23-11516-t003:** Number of m6A-enriched RNAs up- or down-regulated after METTL3 silencing in each cell line (SW1736 or 8505C) and crossing data for the two cell lines (common).

	Up-Regulated	Down-Regulated
SW1736	122	94
8505C	81	33
Common	6	1

**Table 4 ijms-23-11516-t004:** List of m6A genes enriched and commonly up- or down-regulated in the two ATC cell lines.

Gene	Regulation
*ELF2*	Up-regulated
*MUC1*	Up-regulated
*ROM1*	Up-regulated
*TCEANC*	Up-regulated
*THNSL1*	Up-regulated
*ZNF263*	Up-regulated
*DDI2*	Down-regulated

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
