# Peer review of "Role of m6A RNA Methylation in Thyroid Cancer Cell Lines"

_ijms, 2022, doi:10.3390/ijms231911516_

Round 1
Reviewer 1 Report
The authors present the manuscript Role of m6A RNA methylation in thyroid cancer cell lines. The authors indicate a role for m6A RNA methylation in thyroid cancer cell lines by utilizing an siRNA approach.
The authors identify DDI2 asa target gene of interest. Since N6-methyladenosine (m6A) is the most abundant modification with an average of 1–2 m6A residues every 1000 nucleotides, it is difficult to understand genetic regulatory networks solely by siRNA approaches.
The reviewer appreciates the authors use of multiple cell lines to verify datasets regarding methylation profiles of thyroid cancers.
Could the authors indicate the difference between Nthy-ori-3.1 and 8505C cells as compared to SW1736 as compared to BCPAP cells, as there seems to be three distinct clusters on PCA
Along those same lines of thought. What processes are uniquely different between Nthy-ori-3.1 and 8505C cells and SW1736 cells. (Figure 2). For instance since these cells segregate on PCA, then the top ten pathways altered by methylation status should be somewhat different, no?
Table 3 seems to be unclear and irrelevant.
IN figure 3, have the authors attempted a pool of siRNA that would result in greater knockdown of METTL3. Also the authors should perform an additional assay besides MTT so as to confirm alterations in cell viability.
In figure4, have the authors developed a clear rationale for why 8505 cells have little alteration post siRNA treatment?
In figure 6, The reviewer kindly suggests to used a pooled approach, or try more siRNAs against DDI2. The knockdown effect appears weak, yet the actin blots look difficult to quantify.
have the authors performed MTT assays post siDDI2 treatment?
--------------------
The authors present the manuscript Role of m6A RNA methylation in thyroid cancer cell lines. The authors indicate a role for m6A RNA methylation in thyroid cancer cell lines by utilizing an siRNA approach.
The authors identify DDI2 asa target gene of interest. Since N6-methyladenosine (m6A) is the most abundant modification with an average of 1–2 m6A residues every 1000 nucleotides, it is difficult to understand genetic regulatory networks solely by siRNA approaches.
The reviewer appreciates the authors use of multiple cell lines to verify datasets regarding methylation profiles of thyroid cancers.
Could the authors indicate the difference between Nthy-ori-3.1 and 8505C cells as compared to SW1736 as compared to BCPAP cells, as there seems to be three distinct clusters on PCA
Along those same lines of thought. What processes are uniquely different between Nthy-ori-3.1 and 8505C cells and SW1736 cells. (Figure 2). For instance since these cells segregate on PCA, then the top ten pathways altered by methylation status should be somewhat different, no?
Table 3 seems to be unclear and irrelevant.
IN figure 3, have the authors attempted a pool of siRNA that would result in greater knockdown of METTL3. Also the authors should perform an additional assay besides MTT so as to confirm alterations in cell viability.
In figure4, have the authors developed a clear rationale for why 8505 cells have little alteration post siRNA treatment?
In figure 6, The reviewer kindly suggests to used a pooled approach, or try more siRNAs against DDI2. The knockdown effect appears weak, yet the actin blots look difficult to quantify.
have the authors performed MTT assays post siDDI2 treatment?
Author Response
Thank you for the comments. In Replying we believe that the manuscript has been greatly improved
- Could the authors indicate the difference between Nthy-ori-3.1 and 8505C cells as compared to SW1736 as compared to BCPAP cells, as there seems to be three distinct clusters on PCA?
REPLY: Nthy-ori-3.1 are normal human primary thyroid follicular epithelial cells immortalized by transfection with a plasmid containing an origin-defective SV40 genome (SV-ori). 8505C cells were established from the undifferentiated thyroid carcinoma of a 78 year old female patient; this primary carcinoma tissue contained some residual well differentiated components suggesting well differentiated to undifferentiated carcinoma progression. SW1736 are anaplastic thyroid carcinoma cells isolated from a 77 year old affected patient. BCPAP cells were established from the tumor tissue of a 76-year-old woman with metastatic papillary thyroid carcinoma.
Thus, we can hypothesize that the strong heterogeneity in the observed patterns of m6A could indeed be correlated to the great heterogeneity of the cell lines themselves.
In the revised version, at lines 95-97, it is now stated: ” Taking into account the totality of m6A enriched mRNAs, the principal component analysis (PCA), revealed a clear separation between Nthy-ori-3.1, BCPAP and SW1736 cells (Figure 1, panel a), indicating heterogeneity among the four cell lines. Unexpectedly, Nthy-ori-3.1 and 8505C cells are relatively close together. This could be partially explained by the fact that 8505C retains partial features of a well-differentiated carcinoma, whereas SW1736 are completely de-differentiated; this spatially places them between Nthy-ori-3.1 and SW1736 in the PCA.”
- Along those same lines of thought. What processes are uniquely different between Nthy-ori-3.1 and 8505C cells and SW1736 cells. (Figure 2). For instance since these cells segregate on PCA, then the top ten pathways altered by methylation status should be somewhat different, no?
REPLY: Following to the criticism of the Reviewer, the pathway analysis has been better commented. In the revised version, at lines 122-135 is now stated: “5 pathways are common for all four cell lines (Wnt, CCKR, GNRH signaling, angiogenesis, and apoptosis). Interestingly, though Nthy-ori-3.1 and 8505C cells are close in the PCA, only the TGF-beta signaling pathway is unique and common. Seven pathways are shared between the two anaplastic cell lines.”
- Table 3 seems to be unclear and irrelevant.
REPLY: In accordance with Reviewer suggestion, Table 3 was removed from the text.
- IN figure 3, have the authors attempted a pool of siRNA that would result in greater knockdown of METTL3?. Also the authors should perform an additional assay besides MTT so as to confirm alterations in cell viability.
REPLY: In the early experimental stages, we tested a pool of METTL3 siRNAs acting together, but we did not obtain satisfactory results in terms of reducing protein levels of METTL3. We, therefore, preferred to continue the study with three different siRNAs directed against three different sequences of the METTL3 mRNA, then focusing on the two that showed the greatest reduction. Although a greater or complete reduction in the protein level of METTL3 would have been a better goal, we consider the levels of downregulation achieved nevertheless satisfactory, both in light of similar work in the literature, (see: Silencing METTL3 inhibits the proliferation and invasion of osteosarcoma by regulating ATAD2, Lei Zhou et al, https://doi.org/10.1016/j.biopha.2020 .109964; Silencing of METTL3 effectively hinders invasion and metastasis of prostate cancer cells, Yabing Chen et al, doi: 10.7150/thno.61178), and because a reduction in methylation levels is achieved, which is the desired goal.
Unfortunately, we are unable to perform cell viability assays other than the MTT assay, however, cells after each treatment were counted using trypan blue. We have added these results in the supplementary data. We are confident that this may be an additional method of assessing cell viability/proliferation.
- In figure 4, have the authors developed a clear rationale for why 8505 cells have little alteration post siRNA treatment?
REPLY: The reason why 8505C cells show reduced alteration after treatment with siRNAs directed against METTL3 will surely need more in-depth studies, which could be the subject of future investigations. However, at the moment, the most consistent hypothesis concerns the fact that this cell line exhibited extremely low m6A enrichment. It suggests that given the small number of mRNAs bearing such an alteration, switching off their methylator would changes only the small proportion of methylated mRNAs. Therefore, in the revised version, at lines 295-299, this concept was commented.
- In figure 6, The reviewer kindly suggests to used a pooled approach, or try more siRNAs against DDI2. The knockdown effect appears weak, yet the actin blots look difficult to quantify. Have the authors performed MTT assays post siDDI2 treatment?
REPLY: Evaluating DDI2 silencing to assess its role in anaplastic thyroid cancer is beyond the scope of this study, although it could represent an important starting point for future studies. Figure 6 shows the effects of the two siRNAs directed against METTL3 on DDI2 mRNA (A) and protein (B and C) levels. Indeed, the reduction of METTL3 levels affects the expression of DDI2, supporting the hypothesis that methylation of that mRNA (DDI2) is required for its expression.
Reviewer 2 Report
Very good and interestng artice
Author Response
Thanks for your review
Reviewer 3 Report
The authors compared m6A-related RNA methylation profiles between a normal thyroid cell line and three thyroid cancer cell lines, and identified DDI2 as a gene specifically controlled by the m6A modification.
Presented results, although preliminary, and conducted only in continuous cell models, warrant further search for new molecular patterns for diagnostic discrimination between benign and malignant lesions, and the study of the role of m6A in thyroid cancer.
Overall it is well written and discussed paper, however some minor improvements are necessary:
- tables should be reformatted to assure better readability,
- spacing in lines 236, 305, and 440 should be corrected
- number 2 in line 321 should be in subscript
- section 4.9 –sequences of the primers should be provided in the supplementary material.
Author Response
Thank you for the comments that have been all addressed
- tables should be reformatted to assure better readability
REPLY: We experienced formatting problem creating the final PDF in the MDPI portal. We will make sure that this will not happen in the final version of our revised manuscript.
- spacing in lines 236, 305, and 440 should be corrected
REPLY: Spacing has been corrected.
- number 2 in line 321 should be in subscript
REPLY: The suggested change has been made.
- section 4.9 –sequences of the primers should be provided in the supplementary material.
REPLY: The sequences of the primers used in this study are now available as supplementary data.